# Protocol for development and validation of instruments to measure women's empowerment in urban sanitation across countries in South Asia and Sub-Saharan Africa: the Agency, Resources and Institutional Structures for Sanitation-related Empowerment (ARISE) scales

Sheela S Sinharoy [iD],[1] Amelia Conrad,[2] Madeleine Patrick,[2] Shauna McManus,[3] Bethany A Caruso [iD] [1]

For numbered affiliations see end of article.

**Correspondence to**
Dr Sheela S Sinharoy;
sheela.sinharoy@emory.edu

## ABSTRACT

**Introduction** Despite an increasing emphasis on gender and empowerment in water, sanitation and hygiene (WaSH) programmes, no rigorously validated survey instruments exist for measuring empowerment within the WaSH sector. Our objective is to develop and validate quantitative survey instruments to measure women's empowerment in relation to sanitation in urban areas of low-income and middle-income countries.

**Methods and analysis** We are developing the Agency, Resources and Institutional Structures for Sanitation-related Empowerment scales through a process that involves three phases: item development; scale development and initial validation and scale evaluation and further validation. The first phase includes domain specification, item generation, face validity and content validity assessment and item refinement. The second phase involves a second round of face validity and content validity assessment, followed by survey implementation in two cities (Tiruchirappalli, India and Kampala, Uganda) and data analysis involving factor analysis and item response theory approaches as well as reliability and validity testing. The third phase involves a final round of face validity and content validity assessment, followed by survey implementation in three additional cities (Narsapur and Warangal, India and Lusaka, Zambia) and statistical analysis using similar approaches as in phase 2 for further validation.

**Ethics and dissemination** Ethics approvals have been received from the Emory University Institutional Review Board (USA); Azim Premji University and Indian Institute of Health Management Research Institutional Review Boards (India); Makerere University School of Health Sciences Research and Ethics Committee (Uganda); and ERES Converge Institutional Review Board (Zambia). The study team will share findings with key stakeholders to inform programming activities and will publish results in peer-reviewed journals.

### Strengths and limitations of this study

► We employ rigorous methods to develop survey instruments, including testing and validating 15 scales, to assess domains of women's empowerment related to sanitation.
► Phased data collection is being carried out across five urban locations in South Asia and Africa to ensure that the survey instruments are valid and comparable across contexts.
► Our survey instruments are specific to women in urban settings; adaptation and further validity testing are needed to develop instruments that can be administered in other settings (eg, rural, periurban) and with men.
► The survey instruments are focused on empowerment at the individual, household and community levels; they do not assess empowerment in markets (eg, across the sanitation value chain) or in relation to policies and formal governance.

## INTRODUCTION

Despite significant investment and prioritisation, improved water and sanitation access remain out of reach for large portions of the global population. Only 71% of the global population has access to safely managed water and only 45% has access to safely managed sanitation.[1] Water, sanitation and hygiene (WaSH) programmes have postulated that improving water and sanitation access will

lead to improved health, well-being and empowerment of women.[2–4] However, consistent measurement of empowerment outcomes of WaSH programmes remains lacking.

A number of definitions, conceptualisations and measurement approaches have been proposed for women's empowerment. Perhaps the most commonly used definition is that of Kabeer, who described empowerment as 'the expansion in people's ability to make strategic life choices in a context, where this ability was previously denied to them'.[5] This definition emphasises that empowerment is a process, which has led some researchers to argue that empowerment is best measured qualitatively.[6] However, consensus on measurement of empowerment is lacking. As stated in one review, 'this field of study still lacks a coherent conceptual framework for measurement that can guide researchers in how to operationalise empowerment'.[7] A number of indicators and survey instruments have been developed for measuring women's empowerment, either broadly or in specific sectors (eg, agriculture).[8] However, no rigorously validated survey instruments exist for use within the WaSH sector.

To address the lack of validated measures of empowerment in the WaSH sector, we plan to carry out a validation study. Our study objective is to develop and validate quantitative scales to measure domains and subdomains of women's empowerment in relation to sanitation in urban areas of low-income and middle-income countries. We define a scale as a measure in which items' values are determined by an underlying construct.[9] The Agency, Resources and Institutional Structures for Sanitation-related Empowerment (ARISE) scales will assess women's empowerment in relation to urban sanitation across 15 sub-domains of empowerment. The ARISE scales will provide researchers, practitioners and policymakers with tools to improve the design of sanitation programmes and policies, evaluate the impacts of interventions and monitor progress towards global targets, such as the Sustainable Development Goals. The scales will also enable examination of the relationship between domains of sanitation-related women's empowerment and sanitation access, sanitation behaviours and other characteristics, such as caste and area of residence.

## METHODS AND ANALYSIS
Our process for developing the ARISE scales involves three phases: item development; scale development and initial validation and scale evaluation and further validation (see table 1).

### Phase 1: item development
#### Domain specification
The domains and subdomains of the scales are based on the conceptual model of women and girls' empowerment developed by Van Eerdewijk et al.[10] While many conceptual frameworks of empowerment have been developed, this framework has been adopted by the Bill & Melinda

Gates Foundation (BMGF) to inform their programming, including within and beyond WaSH. Van Eerdewijk et al define empowerment as 'the expansion of choice and strengthening of voice through the transformation of power relations, so women and girls have more control over their lives and futures.'[10] The conceptual model includes three domains of empowerment—resources, agency and institutional structures—each with multiple subdomains.[10]

Using this conceptual model, we conducted a systematic review of peer-reviewed literature related to empowerment in WaSH. Details of the methods and results of the systematic review are described elsewhere.[11] We developed sanitation-specific definitions for each subdomain of empowerment, drawing on the conceptual model's definitions with adaptations informed by the literature and team members' expertise (table 2). Adaptations included removing the mention of 'girls' and excluding the market and state subdomains of institutional structures, as well as laws and policies, to focus on adult women's empowerment at individual, household and community levels. Of note, we identified and defined two additional subdomains of empowerment relevant to sanitation from the systematic review: privacy and freedom of movement (within resources and agency, respectively).[11]

### Item generation
Through the systematic review and a parallel landscape analysis, we searched for existing instruments to measure each domain and subdomain of empowerment. As we identified instruments, we collated and assessed them for potential use and/or adaptation in our scales. While the systematic review focused on peer-reviewed publications in WaSH, the landscape analysis included grey literature and was not restricted to the WaSH sector. The landscape analysis identified 62 instruments that had not been identified through the systematic review (online supplemental table 1).

Candidate items were identified or created deductively, through a multistep process.[12] First, we identified broad topics that had emerged through the systematic review for each subdomain of empowerment. We simultaneously compiled items identified in the systematic review and landscape analysis, organising them by empowerment subdomain. Where possible, we revised or adapted existing items to align with identified subdomain-specific topics; most newly developed items were developed to fill gaps in emergent topics that existing items did not meet. Across all subdomains, we allowed for some redundancy, formulating items to capture the same latent construct in different ways. Finally, we reviewed the item sets by subdomain of empowerment, focusing on comprehensiveness and alignment with our operational definitions and made revisions as needed. This resulted in 15 draft scales, one for each subdomain of empowerment. All scales had ordinal response options of 'strongly disagree', 'disagree', 'agree' and 'strongly agree'; frequency response options of 'never', 'sometimes', 'often' and 'always' or, in the case

**Table 1** Overview of planned methods and analyses for the development of the MUSE Scales

| Activity | Procedures |
|---|---|
| *Phase 1: item development* | |
| 1.1 Domain specification | We specified domains and sub-domains based on an existing conceptual model of empowerment and a systematic literature review. |
| 1.2 Item generation | We developed candidate items for each sub-domain through a multi-step process, beginning with a systematic literature review and a parallel landscape analysis. |
| 1.3 Face validity and content validity assessment | We assessed face validity through cognitive interviews (CIs) conducted in two sites (Tiruchirappalli, India and Kampala, Uganda) and content validity through evaluation by four expert reviewers. We also conducted key informant interviews (KIIs) for additional insight. |
| 1.4 Item refinement | We revised items based on field notes from CIs, KIIs and expert feedback, adding, eliminating, combining, and re-phrasing items. |
| *Phase 2: scale development and initial validation* | |
| 2.1 Content validity assessment for new and modified items | We conducted CIs with respondents in Tiruchirappalli and Kampala to assess the refined items. |
| 2.2 Survey participant selection in two sites | We randomly sampled female participants ages 18+in Tiruchirappalli and Kampala in purposively selected neighbourhoods in coordination with BMGF-funded partners and local government. |
| 2.3 Survey implementation and management in two sites | We trained enumerators who carried out tablet-based data collection with approximately 1000 women in each of the two sites. Teams also targeted up to 100 women per site for retesting. |
| 2.4 Statistical analysis | We developed an analysis plan a priori, which included the following steps: 2.4 a) initial item reduction; 2.4b) factor extraction and further item reduction using exploratory factor analysis (EFA); 2.4 c) item reduction using item response theory (IRT); 2.4d) dimensionality confirmation using confirmatory factor analysis (CFA); 2.4e) measurement invariance assessment using multiple-group CFA and IRT; 2.4 f) scale scoring; 2.4 g) reliability testing; and 2.4 h) validity testing. |
| *Phase 3: scale evaluation and further validation* | |
| 3.1 Content validity assessment for new and modified items | We will conduct CIs with respondents in two additional locations to assess any items refined or added based on statistical analyses (2.4). |
| 3.2 Survey participant selection in three sites | We will administer cross-sectional surveys to a validation sample in three cities. The sampling strategy, participant selection strategy, and inclusion criteria are the same as Phase 2. |
| 3.3 Data collection and management | We will conduct enumerator training and survey implementation using tablet-based data collection. |
| 3.4 Statistical analysis | We will conduct EFAs and IRT on data from each site to evaluate any new items generated in Phase 2, and conduct CFAs to test the factor structures identified in Phase two using the same fit indices and criteria to assess model fit. We will test for measurement invariance at both the group and item level using multiple-group CFA and IRT methods; calculate the reliability coefficient; and test for convergent, discriminant, known-groups, and external criterion validity. |

of Freedom of Movement, 'not at all', 'only with accompaniment', 'alone with permission', 'alone if I tell someone' or 'alone without telling anyone'

In addition, we developed six draft indices. An index, in contrast to a scale, does not represent a latent construct. Rather than having a shared cause in the form of an underlying latent construct, the items in an index together share an effect on a latent construct.[9] We developed five indices to measure women's actual experiences in relation to the four subdomains of agency (household-level decision-making, community-level decision-making, leadership, collective action and freedom of movement). The sixth index was for safety and security, to measure women's awareness of actual experiences of sanitation-related violence among women they know. In addition to providing valuable information on women's actual experiences, the indices also contribute to scale validation. We hypothesised that associations would exist between women's self-reported experiences (or awareness of others' experiences) and the corresponding sub-domains

**Table 2** Sanitation-specific definitions for sub-domains of empowerment, by domain

| Sub-domain | Sanitation-specific definition |
| --- | --- |
| **Agency** | |
| Decision-making | Women influence and make decisions about sanitation inside and outside the home. |
| Leadership | Women assume leadership positions, effectively participate and support women's leadership in informal and formal sanitation initiatives and organisations. |
| Collective action | Women gain solidarity and take action collectively on sanitation-related issues. |
| Freedom of movement | Women have the autonomy to move freely to access sanitation facilities, collect water for sanitation-related needs and/or attend forums on sanitation issues, and women have freedom of movement despite sanitation circumstances. |
| **Resources** | |
| Bodily integrity | Women's control over their bodies and ability to access and use their preferred sanitation location. |
| Health | Women's complete physical, mental and social well-being as affected by sanitation options and conditions; not merely the absence of disease or infirmity.[35] |
| Safety and security | Women's freedom from acts or threats of violence (physical or sexual), coercion, harassment, or force when accessing and using sanitation locations or engaging in sanitation-related decision-making processes in the public sphere. |
| Privacy | Women's ability to maintain desired levels of privacy when accessing and using sanitation locations. |
| Critical consciousness | Women's ability to identify and question how inequalities in power operate in their lives in relation to sanitation access and decision-making processes, and to assert and affirm their self-efficacy inside and outside of the household as it relates to sanitation. |
| Financial and productive assets | Women's control over economic resources and long-term stocks of value such as land, for the purposes of meeting individual and household sanitation needs. |
| Time | Women's control over their time and labour spent on sanitation-related tasks and activities. |
| Social capital | Women's relations and social networks that provide tangible and intangible value and support, including those that enable them to complete sanitation-related tasks and activities. |
| Knowledge and skills | Women's knowledge and skills related to sanitation (eg, operation and maintenance of sanitation facilities) and their abilities to apply those knowledge and skills. |
| **Institutional structures** | |
| Norms | Collectively held expectations and beliefs of how women and men should behave and interact inside and outside the household, specifically with regard to sanitation-related (a) division of labour; (b) decision-making; (c) leadership; (d) collective action and (e) freedom of movement. |
| Relations | The interactions and relations—including conflicts, support, hostility and communication—with key actors that shape women's sanitation-related experiences. |

of agency, which would enable the assessment of construct validity of the scales.

### Face validity and content validity assessment

We assessed face validity and content validity (ie, if items adequately measure the domain of interest) of the initial scales through two methods: cognitive interviews (CIs) and expert review.[13] Due to the length of the scales, CI guides were prepared separately for each of the three domains of empowerment (Agency; Resources; and Institutional Structures).

CIs took place in two cities, Tiruchirappalli, India and Kampala, Uganda, in July and August 2019. These cities were selected by BMGF based on the presence of existing BMGF-funded citywide inclusive sanitation (CWIS)

programmes and the strength of local partners. In each city, 12 interviewers were recruited, all with research experience and local language skills. Interviewers received a 5-day training on the purpose of the research, key concepts and definitions of empowerment, CI methods, ethics and all aspects of the CI guides, with time to practice and provide feedback on translations and phrasing.

Neighbourhoods were purposively selected in each city with the goal of sampling from different wealth strata. We defined the study population *a priori* to be adult women (age 18 and older). With the support of local leaders (Uganda) and staff who had previously carried out data collection in these areas (India), the research team purposively sampled participants with the aim of

having an equal distribution across three subgroups to represent varied life stages: between ages 18 and 25 and preferably unmarried; between ages 25 and 40 and preferably married or living with a partner; and over age 40. A city coordinator and/or field supervisor was present throughout data collection to guide participant selection, monitor quality control and provide feedback.

In each city, data collection teams—comprised of one interviewer and one note-taker—conducted CIs with 9–16 individuals per domain of empowerment. Interviews were conducted in the local language (Tamil in Tiruchirappalli and Luganda or English in Kampala). Participants were asked to 'think aloud' about each survey item, and interviewers probed participants to assess understanding and relevance of each item and to identify opportunities for improving wording and translation. Responses were audio recorded and note-takers took detailed field notes. In addition to the women's empowerment items, data collection included questions on sociodemographic characteristics of the respondent and her household as well as on access to and behaviours related to sanitation.

Members of our team led daily debriefings with data collection teams and took detailed field notes on data collectors' reports of scale items that were difficult to administer or understand, confusing or missing response options and other problems. Team members also kept detailed notes throughout the training, data collection and debriefing processes to capture emerging issues and additional topics that should be included or excluded during scale revision.

Key informant interviews (KIIs) were also conducted with WaSH and gender experts in Uganda (n=13) and India (n=20) to provide additional contextual information and insight to inform the inclusion, exclusion or addition of items. Key informants were selected with the assistance of local teams in each site and included representatives from government, non-governmental organisations, the private sector and academia.

Finally, a panel of four expert reviewers evaluated candidate items. Each expert reviewer received the draft scales, our sanitation-specific definitions for each subdomain of empowerment and instructions and background information. They provided item-specific and overall comments and feedback.

### Item refinement and survey preparation for Phase 2

We revised the scales based on field notes from the CIs, KIIs, debriefings and feedback from the expert reviewers. Revisions included eliminating, combining and rephrasing items. We also added new items where gaps had been noted, especially sanitation-specific menstruation items and items about interactions between individuals and sanitation service providers. In preparation for phase 2, we added measures to assess the criterion and construct validity of each scale (online supplemental table 2).

### Phase 2: scale development and initial validation
### Content validity assessment for new and modified items

Following refinement and revision of the scale, beginning in November 2019, trained data collectors (selected from among those who worked on phase 1 of the study) conducted a second round of CIs with 12 individuals each in Tiruchirappalli and Kampala to evaluate newly added items and translations prior to broader survey implementation. Team members carried out rapid analysis of data from the CIs and revised the survey instruments as needed.

### Sample size and participant selection in Tiruchirappalli and Kampala

Following revisions from phase 2 CIs, cross-sectional surveys were administered to a sample of respondents in Tiruchirappalli and Kampala.

Consensus is lacking on optimal sample size for scale development. However, the literature suggests a range of 5–15 respondents per item.[14–16] We aimed for each scale to have at least 15 responses per item. At the time of the calculations, our longest scale consisted of 61 items, requiring a sample size of 915 respondents. We rounded this number up to 1000 respondents per city to allow for approximately 10% non-response and missing data. We aimed to readminister the survey with 5%–10% of participants (N=50–100) per city to assess test–retest reliability.

As in phase 1, we purposively selected neighbourhoods in each city from low and moderate wealth strata, in coordination with CWIS partner organisations and local government. Specifically, partners provided lists of neighbourhoods in each city and identified several priority areas for data collection. In Tiruchirappalli, our team worked with the CWIS partners to select priority slum neighbourhoods and non-priority middle-income neighbourhoods. In Kampala, our team matched priority neighbourhoods to other neighbourhoods of similar population size and income level, using census data.

Participant selection followed a simple random sampling strategy, specifically a random-walk sampling method. Enumerators were instructed to walk in pairs through selected neighbourhoods, with one enumerator working on each side of the street and to knock on every third door. To be eligible, the respondent needed to be a woman age 18 or older who spoke Tamil (in India) or English or Luganda (in Uganda), who was mentally competent (demonstrated through understanding of the study description and consent) and had no hearing or speech impediments that would prevent comprehension or participation. If the selected household had an individual present who met the inclusion criteria and consented to participate, the enumerator would administer the survey.

Respondents were asked whether they would be willing to participate in the same survey a second time. Those who completed the full survey and agreed to participate again were revisited within 1 month. We selected a 1-month time frame because several of the scales refer

to 'the past month,' and we aimed to capture responses within an overlapping period to reduce the likelihood of a meaningful change in any domains of women's empowerment leading to changes in responses to survey items.

### Survey implementation and management in two sites

In each site, 12–14 female enumerators were recruited. As in phase 1, enumerators completed 5 days of training covering key concepts of the study, research ethics and logistics, with 2 days for interactive practice administering the survey to one another and talking through any challenges.

Enumerators first obtained oral (India) or written (Uganda) consent, then administered the full survey instrument. The survey instrument included all 15 scales and six indices, plus modules on demographics, water and sanitation access and behaviours, menstruation and additional items to assess validity. Each site had at least one city coordinator and/or field supervisor who provided day-to-day supervision and quality control. Data collection lasted 24–30 days in each site in December 2019 and January 2020.

Data were collected using tablets with Ona software ( ona.io). In each site, the survey was programmed in both English and either Tamil (India) or Luganda (Uganda). Field supervisors checked surveys for completeness at the end of each day. Data were then uploaded to a secure data storage platform.

### Statistical analysis

Following initial data cleaning and management, we will randomly split the data into equal subsamples for exploratory and confirmatory factor analyses. We will examine summary statistics for items, then follow an a priori analysis plan that includes elements of classical test theory and item response theory (IRT). The analysis plan involves the following steps: item reduction based on missingness, factor extraction, item reduction based on IRT, dimensionality confirmation, measurement invariance assessment, scale scoring, reliability testing and validity testing. Factor analysis will be conducted using Mplus software; all other analyses will be conducted using SAS and R.

#### Initial item reduction

First, we will drop any items with cumulative missingness >30% (including those for which respondents answered 'don't know' or 'not applicable'), with the exception of items related to menstruation. We anticipate that our sample will include women who are not menstruating for many reasons, including pregnancy, lactation and not being of reproductive age.

#### Factor extraction and further item reduction

We will use the first random subsample to perform exploratory factor analysis (EFA) for factor extraction. EFA is appropriate because a factor structure has not been previously established for the items.[17] To determine how many factors to extract, we will use parallel analysis and scree plots.[18]

We will run EFA models on each scale, using means-adjusted and variance-adjusted weighted least square estimators, which are appropriate for items with ordinal response options.[19] We will use oblique rotation to allow for correlation between the factors representing subdomains of empowerment.[17 18] We will examine model statistics from several types of oblique rotations to assess differences across methods and to choose an oblique rotation method. We will interpret model fit based on the following indices: root mean squared error of approximation (RMSEA), comparative fit index (CFI), Tucker-Lewis Index (TLI) and standardised root mean squared residual (SRMR). RMSEA <0.08, CFI >0.95, TLI >0.95 and SRMR <0.06 are considered good fit.[20]

After running each model, we will drop items based on low pattern coefficients (ie, <|0.300|), high multidimensionality (ie, cross-loadings (>|0.300|) on two or more factors with a difference between loadings of <0.20) or significant negative pattern coefficients. We require a minimum of two items per factor and will remove items that load alone on a factor.

#### Item reduction using IRT

In addition to factor analysis, we will use IRT approaches to assess psychometric properties of individual items within each scale. We will use graded response models (GRM), which are appropriate for items with ordinal response options.[21] We will evaluate the assumptions of local independence and functional form and assess model data fit visually and statistically.[21 22] Items identified as having local dependence will be considered for removal, as will items with poor item-level model data fit.[21] We will examine item properties, including item information curves (IIC) and option characteristic curves (OCC), also known in GRMs as item response category characteristic curves.[21 23–25] We will calculate slope (discrimination) and threshold (difficulty) parameter estimates and visually examine IIC and OCC plots to assess item performance.[23–25] We plan to iteratively drop items and run the analysis with the remaining variables until a final model is reached.

#### Dimensionality confirmation

We will use confirmatory factor analysis (CFA) on the remaining random split-half sample to test the structures that were identified through the above process. We will use the same fit indices and criteria described above to assess model fit.

#### Scale scoring

We will calculate sums and mean scores for each scale and use both unweighted (sum and mean) and weighted (factor scores) scores for subsequent validity assessments.

#### Reliability testing

To assess internal consistency, we will examine inter-item correlations and calculate the reliability coefficient, coefficient omega, which, unlike coefficient alpha, does not assume equal covariances of items with their common

factor.[26][27] While consensus is lacking on thresholds for values of omega, we determined that 0.70 would be acceptable.[28][29]

In addition, we will assess test–retest reliability using data from the subsample of respondents who completed the survey two times within a 1-month period. We assume that respondents' level of empowerment would remain stable during this timeframe and that changes in responses would only reflect the stability of our measure. We will estimate test–retest reliability by calculating intraclass correlation coefficients (ICCs) of scored scales. ICCs will be calculated with two-way mixed effect models of absolute agreement of the mean of k items.[30]

### *Validity testing*
We will test for construct validity and external criterion validity using non-parametric spearman-rank correlations and generalised linear regression, with items added for this purpose as described above and shown in online supplemental table 2. We will use t tests and analysis of variance (ANOVA) to test for known-groups validity. We will also examine inter-item and item-scale correlations and consider dropping items with low values.

On completion of statistical analysis, we will make final decisions about items to be included in the scales, taking results for measurement invariance, reliability and validity into account. If revisions are deemed necessary based on the results of the statistical analyses, we will generate new items and/or revise items as needed.

### *Measurement invariance assessment*
We will test for measurement invariance at both the group and item level, including assessment of whether the scales are measuring equivalent constructs, and whether items in the scale have equivalent relationships to those constructs, across populations. We will use multiple-group CFA to test for configural, metric and scalar invariance across groups, comparing data from India and Uganda.[31] We will then use IRT to test for uniform and non-uniform differential item functioning, examining results both visually and statistically.[25][32]

### Phase 3: scale evaluation and further validation
#### Content validity assessment for new and modified items
As in phase 2, trained data collectors will conduct CIs to evaluate newly added items and translations prior to broader survey implementation. CIs will be done in two cities, to be chosen based on data collection schedules. Team members will do a rapid analysis of data from the CIs and revise the survey instruments as needed.

#### Survey participant selection in three sites
We will administer cross-sectional surveys to a sample of respondents in three cities: Narsapur and Warangal (India) and Lusaka (Zambia). As in prior phases, these cities were selected based on the presence of existing CWIS programmes. Data collection is scheduled to take place first in Narsapur and Warangal (August–October 2021), followed by Lusaka (October–December 2021).

For phase 3, we anticipate having fewer items in the scales and, correspondingly, a smaller sample size. In general, a minimum sample size of 300 for EFA and 300 for CFA is considered good.[33] Thus, we will aim for a sample size of 700 respondents per city in phase 3 to ensure a sufficient sample size for EFA and CFA analyses. The strategy for neighbourhood and participant selection, as well as inclusion criteria, will remain the same as in phase 2. Surveys will be implemented in the predominant local language(s).

### Data collection and management
As in previous phases, enumerators will be recruited and given a 5-day training. Data collection will follow the same procedures as in phase 2, with enumerators obtaining consent, administering the full survey instrument in the local language(s) most appropriate to each location and entering responses electronically on a tablet. As in phase 2, the survey instrument will include all 15 scales and six indices, plus modules on demographics, water and sanitation access and behaviours, menstruation and additional items to assess validity. In addition, we will assess mental health using the Center for Epidemiologic Studies Depressions Scale-10 and the Patient Health Questionnaire-4, well-being using the WHO Well-Being Index (WHO-5) and life satisfaction using the Personal Wellbeing Index. Finally, we will add survey items related to COVID-19 to assess how sanitation-related behaviours and experiences may have been impacted by the pandemic.

### Statistical analysis
For any scales that have been substantially revised based on results from phase 2, we will conduct EFA to assess the factor structure, followed by CFA. Following the same procedures as phase 2, we will use GRM IRT approaches to assess psychometric properties of the new and revised items within these scales. For all scales, we will then conduct CFAs to test the optimal factor structures identified in phase 2. We will use the same fit indices and criteria described above to assess model fit. As in phase 2, we will test for measurement invariance at both the group (city) and item level using multiple-group CFA and IRT methods; calculate the reliability coefficient, coefficient omega and test for convergent, discriminant and external criterion validity.

## PUBLIC INVOLVEMENT
Public involvement in the ARISE scale development process occurs in every phase. While the conceptual model for the scales was determined a priori, and survey items were generated deductively, CIs are conducted in every phase to elicit the emic perspective of women in each research context. Participants in the CIs provide item-specific and overall comments and feedback, which leads to survey revision and refinement. This helps to ensure that the scales are reflective of participants' own priorities and experiences. In addition, the CWIS partners

include public providers of sanitation services, such as national sanitation offices and city authorities. These key public stakeholders are invited to provide input on the survey and are involved in the development of a sampling strategy in each city to ensure that study results will be useful to them. Finally, the CWIS partners will also be asked for input on dissemination plans.

## LIMITATIONS

Our survey instruments are specific to women in urban settings; adaptation and further validity testing are needed to develop instruments that can be administered with men and in periurban and rural settings. In addition, the survey instruments are focused on empowerment at the individual, household and community levels. Therefore, they do not assess empowerment in markets (eg, across the sanitation value chain) or in relation to policies and formal governance. Methodologically, best practices include using inductive methods (eg, exploratory qualitative research) to guide item development, employing social network methods to assess social capital, and measuring time use (including in different seasons) as part of measuring empowerment related to time.[16 34] Due to time and budget constraints and concerns about participant burden, we were not able to incorporate these recommended methods into our work.

## ETHICS AND DISSEMINATION
### Ethics

All participants provide oral or written consent to enumerators in their local language using a standardised script. Study activities in phase 2 were reviewed and approved by Institutional Review Boards (IRBs reference number 2019/SOD/Faculty/5.1) and Makerere University (Uganda; reference number 2019–038). Study activities for phase 3 have been reviewed and approved by IRBs at Emory University (USA; IRB 00110271), the Indian Institute of Health Management and Research (India; IRB number IRB/2020–2021/001) and ERES Converge (Zambia; reference number 3 October 2020).

### Dissemination

The study team will share findings with CWIS partners and stakeholders to inform programming activities and will publish results in peer-reviewed journals. We will create training materials, webinars and other guidance documents to support future use of the instruments. Data will be made open access via a data repository per BMGF guidelines.

**Author affiliations**
[1]Hubert Department of Global Health, Rollins School of Public Health, Emory University, Atlanta, Georgia, USA
[2]Gangarosa Department of Environmental Health, Rollins School of Public Health, Emory University, Atlanta, Georgia, USA
[3]Biostatistics and Bioinformatics Department, Rollins School of Public Health, Emory University, Atlanta, Georgia, USA

**Acknowledgements** We are grateful to Deepa Karthykeyan, Kun Zhang, Arjun Sharma, Jacinta Ngabo, Ritah Kobusingye and Josephine Kalenda Goma of Athena Infonomics, as well as Vinod Ramanarayanan and Srishty Arun of Civic Fulcrum, for leading study activities in each country, to all members of the data collection teams, and to the CWIS partners for their input and support. We wish to thank our panel of expert reviewers (listed in alphabetical order): Jenala Chipungu (Centre for Infectious Disease Research, Zambia); Mary Kincaid (IRIS Group); Arundati Muralidharan (WaterAid India); and Lucero Quiroga (Stanford University). We thank CS Sharada Prasad and Ajilé Owens for additional project contributions.

**Contributors** SSS and BAC conceptualised the study and obtained funding. All authors contributed to developing ARISE scale items. SSS and AC wrote the manuscript. AC and MP trained data collection teams and oversaw data collection for Phases 1 and 2. SM developed the data analysis plan, with input from SSS. All authors critically reviewed and approved the final draft of the manuscript.

**Funding** This work was supported by the Bill & Melinda Gates Foundation grant number OPP1191625.

**Competing interests** None declared.

**Patient consent for publication** Not applicable.

**Provenance and peer review** Not commissioned; externally peer reviewed.

**ORCID iDs**
Sheela S Sinharoy http://orcid.org/0000-0003-3077-3824
Bethany A Caruso http://orcid.org/0000-0001-9738-9857

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
