## [Reviewer comments · BMJ Open]

ARTICLE DETAILS

TITLE (PROVISIONAL)	Protocol for development and validation of instruments to measure women's empowerment in urban sanitation across countries in South Asia and Sub-Saharan Africa: The Agency, Resources, and Institutional Structures for Sanitation-related Empowerment (ARISE) Scales
AUTHORS	Sinharoy, Sheela; Conrad, Amelia; Patrick, Madeleine; McManus, Shauna; Caruso, Bethany

VERSION 1 – REVIEW

REVIEWER	KC, Heera Purbanchal University
REVIEW RETURNED	10-Jul-2021

GENERAL COMMENTS	I went through the tool validation protocol. Researchers have very cautiously used different statistical tools. Phase wise manner, standard tools validation process and multiple setting of different countries are the remarkable features for this tool validation. Observing women's empowerment in WASH sector actually will help to know the women's status. This tools after validation would be of great use. I appreciate the researchers for this outstanding contribution.
---

REVIEWER	Gonçalves, Ana Universidade Federal do Rio Grande do Norte
REVIEW RETURNED	15-Jul-2021

GENERAL COMMENTS	We appreciate the opportunity to collaborate with this prestigious journal reviewing the Manuscript BMJ open-2021-053104 entitled "Development and validation protocol for an instrument to measure women's empowerment in urban sanitation across countries: The Agency, Resources, and Institutional Structures for Sanitation-related Empowerment (ARISE) Scales" After reading the article and evaluating the paper personally, we feel that there is some sloppy proofreading of the manuscript: Firstly, the title does not make it clear where the study is carried out. The title uses only the term "across countries". Throughout the text, the author talks about five urban locations in South Asia and Africa. That should be clear in the title. The purpose of the study is not clear. The study is a protocol, but data collection is already in progress. There is no information about the start date or the expected completion date. Additionally, It is not clear what the study design is. For this reason, we were unable to identify which instrument should be adopted for writing the protocol.
---

	There is no sample calculation or reason to justify the number of the adopted population. The very long and confusing methodology prevents us from recommending it for publication. The text is confused; the reading is tiring, the impression is that the article does not seem to be directed to the health area.
--	---

REVIEWER	Khatiwada, Januka International University of Health and Welfare
REVIEW RETURNED	20-Jul-2021

GENERAL COMMENTS	Thank you for providing me with this opportunity to review the protocol. It is praiseworthy that this team is developing and validating the quantitative study tools on WaSH related field, which was lacking till the date. Overall, the protocol is consistent, and methodologically sound. However, some points need more explanation and clarification as follows: 1) First of all the concept of women`s empowerment itself is controversial concept. The authors have mentioned about the definition and the concept of women`s empowerment that they followed in methodology section. It would have been better if authors briefly explained about the gap and rationality of the concept of women`s empowerment referring to the existing evidences and knowledge in introduction section. 2) Please indicate the expected timeline for the phase 3 project.
--

VERSION 1 – AUTHOR RESPONSE

Reviewer: 1

Dr. Heera KC, Purbanchal University

Comments to the Author:

I went through the tool validation protocol. Researchers have very cautiously used different statistical tools. Phase wise manner, standard tools validation process and multiple setting of different countries are the remarkable features for this tool validation. Observing women's empowerment in WASH sector actually will help to know the women's status. This tools after validation would be of great use. I appreciate the researchers for this outstanding contribution.

Reviewer: 2

Dr. Ana Gonçalves, Universidade Federal do Rio Grande do Norte

Comments to the Author:

We appreciate the opportunity to collaborate with this prestigious journal reviewing the Manuscript BMJ open-2021-053104 entitled "Development and validation protocol for an instrument to measure women's empowerment in urban sanitation across countries: The Agency, Resources, and Institutional Structures for Sanitation-related Empowerment (ARISE) Scales"

After reading the article and evaluating the paper personally, we feel that there is some sloppy proofreading of the manuscript:

Firstly, the title does not make it clear where the study is carried out. The title uses only the term "across countries". Throughout the text, the author talks about five urban locations in South Asia and Africa. That should be clear in the title.

Response: We have added “South Asia and Sub-Saharan Africa” to the title as suggested.

The purpose of the study is not clear.

Response: We have added a paragraph to the Introduction to provide additional context, which we hope will clarify the rationale and purpose of the study.

The study is a protocol, but data collection is already in progress. There is no information about the start date or the expected completion date.

Response: Information on the timeline was included in sections 1.3, 2.1, and 2.3. We have added text in Section 3.2 to clarify the study timeline, including that we expect to complete data collection by December 2021.

Additionally, It is not clear what the study design is. For this reason, we were unable to identify which instrument should be adopted for writing the protocol.

Response: We have added a sentence in the Introduction section to explicitly state that this is a validation study.

There is no sample calculation or reason to justify the number of the adopted population.

Response: The sample calculation for Phase 2 is described in Section 2.2, “Sample size and participant selection in Tiruchirappalli and Kampala”. Sample size calculation for Phase 3 is described in Section 3.2, “Survey participant selection in three sites.” We have revised and added text to the latter section (for Phase 3) to provide additional details.

The very long and confusing methodology prevents us from recommending it for publication. The text is confused; the reading is tiring, the impression is that the article does not seem to be directed to the health area.

Response: The methodology of our study is in line with best practices for scale development and validation, which indeed is a long process with many steps, some of which require advanced statistical analysis. This process is described in more detail in some of the works cited in our manuscript, such as the DeVellis textbook (Scale Development: Theory and Applications) and the Boateng et al. 2018 article (Best practices for developing and validating scales for health, social, and behavioral research: a primer). Our process is also similar to other papers published in BMJ Open, such as Young et al. 2019 (Development and validation protocol for an instrument to measure household water insecurity across cultures and ecologies: the Household Water InSecurity Experiences (HWISE) Scale).

Reviewer: 3

Dr. Januka Khatiwada, International University of Health and Welfare

Comments to the Author:

Thank you for providing me with this opportunity to review the protocol. It is praiseworthy that this team is developing and validating the quantitative study tools on WaSH related field, which was lacking till the date.

Overall, the protocol is consistent, and methodologically sound. However, some points need more explanation and clarification as follows:

1) First of all the concept of women`s empowerment itself is controversial concept. The authors have mentioned about the definition and the concept of women`s empowerment that they followed in methodology section. It would have been better if authors briefly explained about the gap and rationality of the concept of women`s empowerment referring to the existing evidences and knowledge in introduction section.

Response: We agree. We have added a paragraph to the Introduction to include these points related to the definition of women’s empowerment, gaps in measurement of empowerment, and rationale for our study. We have also added text to section 1.1 to further explain the rationale behind our conceptualization of empowerment.

2) Please indicate the expected timeline for the phase 3 project.

Response: We have added text to clarify the study timeline.

VERSION 2 – REVIEW

REVIEWER	Gonçalves, Ana Universidade Federal do Rio Grande do Norte
REVIEW RETURNED	27-Sep-2021

GENERAL COMMENTS	We appreciate the opportunity to collaborate with this prestigious journal reviewing the Manuscript BMJ open-2021-053104 entitled "Development and validation protocol for an instrument to measure women's empowerment in urban sanitation across countries: The Agency, Resources, and Institutional Structures for Sanitation-related Empowerment (ARISE) Scales." After reading the article and evaluating the paper personally, we feel that its writing has improved significantly since its last version. However, some points are still worrying: The study aims to develop and validate quantitative scales to measure domains and sub-domains of women's empowerment concerning sanitation in urban areas of low- and middle-income countries across South Asia and Sub-Saharan Africa. In the latter context, I'm not sure if they are countries with comparable social, economic, and cultural realities so that it is possible to use the same instrument for different scenarios. However, I am not sure if they are countries with comparable social, economic, and cultural realities so that it is possible to use the same instrument for different scenarios. Maybe it would be more appropriate to build two instruments for low- and middle-income countries. Page 9 , line 8, they conducted a systematic review of peer-reviewed literature related to empowerment in WaSH. However, details of the methods and results of the systematic review are still unavailable.
---

VERSION 2 – AUTHOR RESPONSE

Reviewer: 2

Dr. Ana Gonçalves, Universidade Federal do Rio Grande do Norte

Comments to the Author:

We appreciate the opportunity to collaborate with this prestigious journal reviewing the Manuscript BMJ open-2021-053104 entitled "Development and validation protocol for an instrument to measure women's empowerment in urban sanitation across countries: The Agency, Resources, and Institutional Structures for Sanitation-related Empowerment (ARISE) Scales."

After reading the article and evaluating the paper personally, we feel that its writing has improved significantly since its last version. However, some points are still worrying:

The study aims to develop and validate quantitative scales to measure domains and sub-domains of women's empowerment concerning sanitation in urban areas of low- and middle-income countries across South Asia and Sub-Saharan Africa. In the latter context, I'm not sure if they are countries with comparable social, economic, and cultural realities so that it is possible to use the same instrument for different scenarios. However, I am not sure if they are countries with comparable social, economic, and cultural realities so that it is possible to use the same instrument for different scenarios. Maybe it would be more appropriate to build two instruments for low- and middle-income countries.

Response: We agree that it is important to consider whether the scales can be used to measure domains and sub-domains of empowerment in different populations and contexts. We have added text to the Methods and Analysis section (sub-section 2.4h) to clarify that measurement invariance testing allows us to assess whether we are, in fact, measuring the same construct across different populations, and whether results will be comparable across settings. The assessment of measurement invariance (also known as measurement equivalence) is a standard approach that has been used in the development of other survey instruments that have been constructed for use across different settings and contexts (e.g. Young SL, Collins SM, Boateng GO, et al. Development and validation protocol for an instrument to measure household water insecurity across cultures and ecologies: The Household Water InSecurity Experiences (HWISE) Scale. *BMJ open*. 2019 Jan 1;9(1):e023558.)

Page 9 , line 8, they conducted a systematic review of peer-reviewed literature related to empowerment in WaSH. However, details of the methods and results of the systematic review are still unavailable.

Response: Happily, the systematic review is now available as a preprint! We have added a citation accordingly.

VERSION 3 – REVIEW

REVIEWER	Gonçalves, Ana Universidade Federal do Rio Grande do Norte
REVIEW RETURNED	27-Dec-2021
GENERAL COMMENTS	We recommend accepting. We recommend accepting.